# Detection of False Data Injection Attacks in Smart Grids Based on Expectation Maximization

**DOI:** 10.3390/s23031683

**Published:** 2023-02-03

**Authors:** Pengfei Hu, Wengen Gao, Yunfei Li, Minghui Wu, Feng Hua, Lina Qiao

**Affiliations:** 1School of Electrical Engineering, Anhui Polytechnic University, Wuhu 241000, China; 2Key Laboratory of Advanced Perception and Intelligent Control of High-end Equipment, Chinese Ministry of Education, Wuhu 241000, China

**Keywords:** false data injection attacks, statistical learning methods, attack detection, attack location, smart grid

## Abstract

The secure operation of smart grids is closely linked to state estimates that accurately reflect the physical characteristics of the grid. However, well-designed false data injection attacks (FDIAs) can manipulate the process of state estimation by injecting malicious data into the measurement data while bypassing the detection of the security system, ultimately causing the results of state estimation to deviate from secure values. Since FDIAs tampering with the measurement data of some buses will lead to error offset, this paper proposes an attack-detection algorithm based on statistical learning according to the different characteristic parameters of measurement error before and after tampering. In order to detect and classify false data from the measurement data, in this paper, we report the model establishment and estimation of error parameters for the tampered measurement data by combining the the k-means++ algorithm with the expectation maximization (EM) algorithm. At the same time, we located and recorded the bus that the attacker attempted to tamper with. In order to verify the feasibility of the algorithm proposed in this paper, the IEEE 5-bus standard test system and the IEEE 14-bus standard test system were used for simulation analysis. Numerical examples demonstrate that the combined use of the two algorithms can decrease the detection time to less than 0.011883 s and correctly locate the false data with a probability of more than 95%.

## 1. Introduction

The current power system is continuously monitored by an energy management system (EMS), and a supervisory control and data acquisition (SCADA) system us used to maintain normal and secure operating conditions [1]. In particular, the SCADA system in the control center uses state estimators to process the received measurements. The estimator obtains the best estimate of the system’s state by filtering incorrect data. These state estimates are then transmitted to all EMS to control the proper functioning of the physical aspects of the grid, such as the power flow calculation.

The measurements collected by the SCADA system include not only measurement noise due to the limited precision of sensors and communication medium, but also errors due to various problems, such as connecting and calibrating a failed meter. To decrease the effects of noise and error, power system researchers have developed many methods to deal with the measurements during state estimation [2,3]. The basic principle of these methods is to use the redundancy of multiple measurements to identify and eliminate anomalies.

Most of the technologies used to protect grid systems are designed to ensure system reliability, such as preventing random failures. However, more and more attention has been paid to preventing malicious network attacks in the recent proposals for smart grids [4]. The operation and control of smart grids depend on the complex network space of computer, software and communication technology [5]. Since measurement components supported by smart devices, such as smart instruments and sensors, play important roles in confirming the real-time physical states of power systems, they are likely to be targets of attack. These measuring devices widely use Internet-based protocols in communication systems, which are open to external networks and lack of hardware to prevent tampering. In order to promote data sharing, enterprise networks, and even individual users, are allowed to connect to the infrastructure of power grid information [6]. Potential complex malicious attacks increase after these network interfaces are introduced into power systems [7,8,9,10]. Liu et al. [11] indicated in 2009 that a new FDIA could bypass bad data detection (BDD) in current SCADA systems and introduce any errors into state estimation without being detected. Malicious covert data injection of network buses will inevitably have a negative impact on power-system state estimation [12,13]. The injection of these malicious data that deviates state estimates away from security values can directly result in serious social and economic losses, and an attacker can utilize the FDIA to manipulate the electricity price of the electric market [14,15,16], and this attack can even result in regional power shortages [17].

Du et al. [18] proposed a method to extract network parameters from the limited data obtained by phasor measurement units (PMUs) when the network parameters are unknown and then use these parameters to build an AC attack model, finally making the state estimation deviate from the securely value. Most of the classical methods used to construct the attack model focus on tampering measurements, such as the power injected into the bus and the power flow between buses. Liu et al. [19] proposed a method to attack network parameters which reduces the number of attack measurements by coordinating the modifications of parameters and other measurements in the power system. The attack method is still applicable in cases where the topology and line impedance of the network are incomplete. Since it is unrealistic for an attack to modify network parameters directly. Liu et al. [20] proposed a more universally applicable attack model. The concrete approach is to tamper with network parameters indirectly by exploiting the vulnerabilities that exist when the network parameters are incorrectly handled.

Several directions have been taken in the research of detecting FDIAs in smart grids. Although these detection methods differ to varying degrees, they can be broadly classified into two broad categories. Detection methods can be categorized as model-based detection algorithms and data-driven detection algorithms. In response to the situation in which network parameters are attacked, [21] proposed a way to detect network parameter attacks based on the inconsistency of historical data and specified network parameters. However, such methods are no longer applicable in detecting combinatorial attacks. Methods to detect FDIAs using differences in the probability distributions of historical and current measurement data may not be applicable any longer, such as assuming the attack vector is a trapezoidal attack or that spurious data injected do not significantly deviate from the historical trend [22,23,24]. In addition, such a detection method will easy cause false detection when encountering actual events, such as sudden changes in the load or from the generator. To deal with this situation, a method was proposed in [25] to detect FDIAs using the difference in the residual probability distribution between historical measurement data and that of current measurement data. This method still maintains good detection performance when facing trapezoidal attacks and real events. Chen et al. [26] proposed a scheme to detect data before state estimation by using vector autoregression model. This scheme uses vector autoregressive model to predict and classifiers to detect, which improves the detection rate based on the autoregressive model. Saleh et al. [27] proposed a detection method to detect FDIAs that destroy the state estimation of PMUs. The phase lock value (PLV) is used to judge whether the phase changes between buses are consistent. If the phase change was no longer constant, the data for the PMU were considered to have been manipulated; otherwise, data security at PMUs was considered. The above are several model-based detection methods.

Unlike model-based detection algorithms for FDIAs, machine learning, as a data-driven technique, implies a huge dependence on historical data of the system under test. Yu et al. [28] proposed a false data injection attack detection method for AC state estimation. When FDIAs exist, their spatial and temporal data correlations may deviate from the correlations under normal conditions. By using wavelet transforms and deep neural networks to analyze the estimated states in continuous time, the proposed method can effectively detect this inconsistency. Xun et al. [29] proposed an extreme learning machine (ELM)-based one-class and one-network (OCON) framework for detecting FDIAs. In this framework, the subnetwork of the state identification layer in OCON uses the ELM algorithm to accurately classify false data and normal data. Almasabi et al. [30] proposed a new method to detect FDIAs using moving average, correlation and machine learning algorithms. The experiments showed that the proposed method is able to detect the attacked PMUs and its timing issues with a high detection rate. Most existing machine-learning-based detection methods generally assume that the labels of the training data are known, which may not be consistent with common sense. Since real-life FDIAs are generally considered as rare events, it may be challenging to obtain the identity of the compromised data. An et al. [31] proposed the use of unsupervised integrated autoencoders connected to a Gaussian mixture model (GMM) to accommodate multiple domains. Attention-based potential representation and minimum error reconstruction features are utilized in the hidden space of the integrated autoencoder. The expectation maximization (EM) algorithm is used to estimate the sample density in the GMM. When the estimated sample density exceeds the learning threshold obtained in the training phase, the sample is identified as an outlier. Since the EM algorithm has the disadvantage of being sensitive to initial values, excellent initialization parameters are required for the next iterative step of the calculation. To deal with this challenge, we are required to develop an unsupervised detection approach.

This paper proposes a detection and location method for the false data injection attacks in smart grid. FDIAs threaten the management and control of grids by tampering with the measurement data of the smart grid systems. In fact, the attacker adds an unknown deviation to the measurement data of a system to launch an FDIA. Since the presence of unknown attacks generates error bias, there are different characteristic parameters for the measurement error contained by false data and that of normal data. Therefore, we used the k-means++ algorithm and the expectation maximization (EM) algorithm to estimate the corresponding parameters of the measured data to eliminate the data affected by the FDIA, and finally achieved the purpose of attack detection. The main contributions of this paper can be summarized as follows:Since the error models of both measurement vectors and state variables with false data have the characteristics of the Gaussian mixture model (GMM), a false data injection attack detection method based on the k-means++ and expectation maximization (EM) algorithms is proposed.To address the fact that the k-means algorithm is sensitive to the initial clustering centers and affects the convergence efficiency, the k-means++ algorithm is proposed to determine the initial estimated parameters of the GMM in a faster iterative approach.The k-means++ algorithm is used to preprocess the data to solve the problem of EM algorithm being sensitive to initial values. It also decreases the calculation complexity of the EM algorithm, and finally detects and locates false data rapidly according to the classification results.

## 2. System Model

For complex information processing of smart grid, it is necessary to generate corresponding mathematical model according to network topology and data of distribution network [32]. The general linear state equation of voltage and current phasors in the smart grid distribution system is as follows [33]:(1)$$y=\underset{=z}{\underbrace{\mathrm{Hx}}}+E$$
where $y\in C^{m}$ is the original measurement vector of voltage and current phasor; z is the noiseless measurement vector; $x\in C^{n}$ is the vector describing the system state variable; $H\in C^{m\times n}$ is the network topology matrix describing the vicinity of a given working point; $e\in C^{m}$ is the measurement error produced by the sensor, where each component is modeled as an independent homodistributed and obeys a complex Gaussian random variable with a zero mean and variance of $\sigma^{2}$.

Attackers use FDIAs to add attack vectors to the measurement vectors to corrupt the measurements available to the operator. The actual measurements after being attacked are
(2)$$y_{a}=\underset{=z}{\underbrace{\mathrm{Hx}}}+E+a$$
where $a\in C^{m}$ is the attack vector; $y_{a}\in C^{m}$ represents the measurement after being attacked by false data injection.

With the rapid development of synchronous phasor measurement units (PMUs), a smart grid can obtain impeccable phasor measurement values by arranging PMUs on the terminal buses [34]. Using these measurements, the system state variable x can be accurately estimated. However, due to the price factor of PMUs, the device cannot be installed on all transmission buses of the power system, and can only cooperate with other sensors to obtain system measurements. One of the attacks considered under this condition is that during the stable operation of the power system, one of the *N* phasor measurements in the measurement vector y is continuously attacked; that is, a component in the attack vector a is not zero. In the subsequent measurement acquisition process, we determine whether the phasor measurements are replaced with false data by K(K≥1) measurement vectors. To facilitate the calculation, the obtained measurement samples are converted from complex representation to real coordinate representation, and then the actual obtained component of the *i*th phase measurement of the *k*th measurement vector $y_{k}\in R^{N\times 2}$ is represented as
(3)$$y_{i,k}=z_{i,k}+e_{i,k}$$
where $y_{i,k}\in R^{1\times 2}$, $z_{i,k}\in R^{1\times 2}$, $e_{i,k}\in R^{1\times 2}$. The error distribution of the secure phase measurement is represented by $p_{E}^{\left(1\right)}\left(e;\mu_{1},\Sigma_{1}\right)$, and the error distribution of the phase measurement tampered with by the attack is represented by $p_{e}^{\left(2\right)}(e;\mu_{2},\Sigma_{2})$. In addition, the phasor measurement error distributions belong to two-dimensional Gaussian distributions with unknown parameters.

For ease of calculation, the actual obtained model for the phasor measurement sample of *K* measurement vectors is written as
(4)y=Z+E
where $y\in R^{NK\times 2}$, $Z\in R^{NK\times 2}$ and $E\in R^{NK\times 2}$ represent the original measurement, actual measurement and measurement error obtained from *K* measurements for *N* phase measurement units, respectively.
(5)$$y={\left[y_{1,1},\cdots ,y_{1,K},\cdots ,y_{N,1},\cdots ,y_{N,K}\right]}^{T}$$
(6)$$Z={\left[z_{1,1},\cdots ,z_{1,K},\cdots ,z_{N,1},\cdots ,z_{N,K}\right]}^{T}$$
(7)$$E={\left[e_{1,1},\cdots ,e_{1,K},\cdots ,e_{N,1},\cdots ,e_{N,K}\right]}^{T}$$

Power-grid operators generally apply a likelihood ratio test to each measurement to judge whether the measurement is correct. However, there are errors in the measurement data that conform to a Gaussian distribution, and the number of false alarms increases as the number of measurements increases, making it more difficult to detect false data. In this study, we used the method of processing the results of multiple measurements as a set of data. Since interrelated measurement data are linked, the probability of false alarms can be decreased by mathematically determining the relationship between the data. However, the difficulty of this method is also in which calculation method should be used to quickly determine the relationship between the data in the group. An inappropriate method is likely to increase the workload of the detection system and decrease the detection efficiency.

## 3. Attack Detection

### 3.1. Maximum Likelihood Estimation

When all measurements Y are considered as a whole, the corresponding measurement error samples E can be seen as coming from two clusters—one with MK correct phasor measurement samples and the other with (N−M)K attacked tampered phasor measurement samples. Without testing, it is impossible to determine which samples of measurements have been tampered with by FDIAs. The probability distribution of the measurement error e for each measurement y according to the assumed statistics can be represented by a Gaussian mixture model (GMM):(8)$$p\left(e;\theta \right)=\sum_{l=1}^{2}\alpha_{l}p_{e}^{\left(l\right)}\left(e;\mu_{l},\Sigma_{l}\right)$$
where $\alpha_{1}=M/N$ and $\alpha_{2}=\left(N-M\right)/N$ are unknown.

In this paper, we derived the distribution parameters of the measurement error by exploiting the asymptotic property of maximum likelihood estimation (MLE). Knowing about the phase measurements associated with the parameters and the actual values derived from the state variables, the maximum likelihood estimate θ for unknown parameters can be solved by maximizing the log-likelihood function globally. According to the noise model assumed in (Equation 8), the log-likelihood function with parameter vector $\theta ={\left[\alpha_{1},\alpha_{2},\mu_{1},\Sigma_{1},\mu_{2},\Sigma_{2}\right]}^{T}$ can be obtained as
(9)$$\begin{array}{cc} L_{I}\left(\theta ;E\right) & =\ln\left[p\left(E;\theta \right)\right]\\ \; & =\ln\left[\prod_{i=1}^{N}\prod_{k=1}^{K}p\left(e_{i,k};\theta \right)\right]\\ \; & =\sum_{i=1}^{N}\sum_{k=1}^{K}\ln\left[\sum_{l=1}^{2}\alpha_{l}p_{E}^{\left(l\right)}\left(y_{i,k}-z_{i,k};\mu_{l},\Sigma_{l}\right)\right] \end{array}$$

The maximum likelihood estimate ${\hat{\theta }}_{ML}$ was obtained by solving
(10)$$\begin{array}{cc} \arg\max_{\theta }\; & L_{I}\left(\theta ;E\right)\\ \mathrm{subject}\;\mathrm{to}\; & \alpha_{1}>0,\alpha_{2}>0\\ \; & \alpha_{1}+\alpha_{2}=1\\ \; & \mathrm{and}\;\mathrm{constraints}\;\mathrm{on}\;\mu_{l},\Sigma_{l}\left(l=1,2\right) \end{array}$$

Since the cost function in (Equation 10) is too complex, we would like to use a method to decrease the complexity of calculating the MLE. Therefore, we introduce a complete dataset {E,γ}, where
(11)$$\gamma ={\left[\begin{array}{c} \gamma_{1,1,1},\cdots ,\gamma_{1,K,1},\cdots ,\gamma_{N,1,1},\cdots ,\gamma_{N,K,1}\\ \gamma_{1,1,2},\cdots ,\gamma_{1,K,2},\cdots ,\gamma_{N,1,2},\cdots ,\gamma_{N,K,2} \end{array}\right]}^{T}$$
contains 2NK random hidden variables whose values reflect which mixed component the random variable in the measurement error E belongs to. $\gamma_{i,k,l}$ is defined as follows:(12)$$\gamma_{i,k,l}=\left\{\begin{array}{c} 1,\;\mathrm{if}\;e_{i,k}\;\mathrm{belong}\;\mathrm{to}\;p_{e}^{\left(l\right)}\left(e;\mu_{l},\Sigma_{l}\right)\\ 0,\;\mathrm{otherwise} \end{array}\right.$$

With unobserved data $\gamma_{i,k,l}$, the complete data are $\left(e_{i,k},\gamma_{i,k,1},\gamma_{i,k,2}\right)$. More specifically, if $e_{i,k}$ is the measurement error of the security data, then $e_{i,k}$ belongs to the first mixture component $p_{e}^{\left(1\right)}\left(e;\mu_{1},\Sigma_{1}\right)$ of the Gaussian mixture model, and its complete data are $\left(e_{i,k},1,0\right)$. If $e_{i,k}$ is the measurement error of the false data, then $e_{i,k}$ belongs to the other components of the Gaussian mixture model, denoted as $\left(e_{i,k},0,1\right)$. The log-likelihood function for complete data is
(13)$$\begin{array}{c} L_{C}\left(\theta ;E,\gamma \right)=\ln\left[p\left(E,\gamma ;\theta \right)\right]\\ =\ln\left[\prod_{i=1}^{N}\prod_{k=1}^{K}p\left(e_{i,k},\gamma_{i,k,1},\gamma_{i,k,2};\theta \right)\right]\\ =\ln\left\{\prod_{j=1}^{N}\prod_{k=1}^{K}\prod_{l=1}^{2}{\left[\alpha_{l}p_{e}^{\left(l\right)}\left(e_{i,k};\mu_{l},\Sigma_{l}\right)\right]}^{\gamma_{i,k,l}}\right\}\\ =\sum_{i=1}^{N}\sum_{k=1}^{K}\sum_{l=1}^{2}\gamma_{i,k,l}\ln\left[\alpha_{l}p_{e}^{\left(l\right)}\left(y_{i,k}-z_{i,k};\mu_{l},\Sigma_{l}\right)\right] \end{array}$$

To avoid ambiguity, the original log-likelihood function $L_{I}\left(\theta ;E\right)$ in (Equation 9) is referred to as the log-likelihood function for incomplete data. Clearly, the newly introduced log-likelihood function $L_{C}\left(\theta ;E,\gamma \right)$ for complete data is much simpler to calculate. For GMM-compliant measurements, the EM algorithm can be used to approximate MLE [35].

### 3.2. K-Means++ Algorithm

Since the EM algorithm has the disadvantage of being sensitive to initial values, the parameter θ needs to be initialized in order to proceed to the next iteration of the calculation. The convergence efficiency is greatly decreased by the randomly chosen initial estimated parameter $\theta^{\left(0\right)}$ due to the information uncertainty in estimating parameter θ. At the same time, whether to get a global optimal solution is also worth considering. The k-means algorithm classifies data according to the minimum distance criterion, which is commonly used in the clustering of data streams; its advantages are simplicity and rapidity [36]. The k-means++ algorithm determines the initial estimated parameters of the Gaussian mixture model with faster iterating than the k-means algorithm. At the same time, the k-means++ algorithm decreases the sensitivity to the initial clustering center, thereby accelerating the rate of convergence.

The idea of the k-means++ algorithm can be summarized in two steps. In the first step, the only difference between k-means++ and k-means algorithms is that the k-means++ algorithm chooses initial clustering centers that are far away from each other rather than randomly. Therefore, the above characteristics allow the k-means++ algorithm to have faster calculation speed. In the second step, sample points in the dataset are assigned to cluster centers that are nearest to each other to form different clusters and recalculate cluster centers.

In this paper, the workflow of k-means++ algorithm can be summarized as three steps.

The first step is to select the initial cluster center. First, a sample e is randomly selected from the data set E as the initial clustering center $c_{1}^{\left(0\right)}$. Then, the Euclidean distance between each sample $e_{i,k}$ and the currently existing clustering center $c_{1}^{\left(0\right)}$ is calculated and denoted by $D(e_{i,k})$. Next, the probability of each sample being selected as the next cluster center is calculated by using
(14)$$p_{c}\left(e_{i,k}\right)=\frac{D{\left(e_{i,k}\right)}^{2}}{\sum_{i=1}^{N}\sum_{k=1}^{K}D{\left(e_{i,k}\right)}^{2}}$$

Finally, the second initial cluster center $c_{2}^{\left(0\right)}$ is selected according to the roulette wheel selection.

The second step is to assign the dataset. Assign each sample of the dataset to the appropriate cluster center according to the principle of minimum Euclidean distance.
(15)$$\gamma_{i,k,l}^{\left(n\right)}=\left\{\begin{array}{c} 1,l=\arg\min_{l}∥e_{i,k}-c_{l}^{\left(n\right)}∥\\ 0,\mathrm{otherwise} \end{array}\right.$$
where (Equation 15) indicates that $e_{i,k}$ belongs to the $c_{l}^{\left(n\right)}$-centered clustering domain.

The third step is to update the clustering centers. At the (n+1)th iteration, the cluster centers of the dataset are recalculated based on the hidden variable $\gamma {}^{\left(n+1\right)}$. The newly calculated cluster centers are then used as the center of mass of the samples belonging to that category.
(16)$$c_{l}^{\left(n+1\right)}=\frac{\sum_{\gamma_{i,k,l}^{\left(n\right)}=1}e_{i,k}}{\sum_{i=1}^{N}\sum_{k=1}^{K}\gamma_{i,k,l}^{\left(n\right)}}$$

### 3.3. EM Algorithm

The idea of EM algorithm is to estimate unknown parameters through two iterations: an expectation (E) step and a maximization (M) step. In the first step (E-step), the conditional expectation of the log-likelihood function for complete data is calculated based on the conditional probability of the hidden variable. In the second step (M-step), the conditional expectation obtained by the E-step is maximized for the desired parameters. Using the estimated parameter θ obtained with the k-means++ algorithm, we proposed the workflow of the EM algorithm for the (η+1)th iteration thereafter.

Step 1 (E-step): The conditional expectation for defining the log-likelihood function of complete data is as follows:(17)$$\begin{array}{cc} Q\left(\theta ,\theta^{\left(\eta \right)}\right) & =E\left\{\ln\left[p\left(E,\gamma ;\theta \right)\right];E,\theta^{\left(\eta \right)}\right\}\\ \; & =\sum_{\gamma }\ln\left[p\left(E,\gamma ;\theta \right)\right]\mathrm{Pr}\left(\gamma |E;\theta^{\left(\eta \right)}\right)\\ \; & =\sum_{\gamma }\ln\left[p\left(E,\gamma ;\theta \right)\right]{\hat{\gamma }}_{i,k,l}^{\left(\eta \right)} \end{array}$$
where ${\hat{\gamma }}_{i,k,l}^{\left(\eta \right)}$ is a shorthand form of the conditional probability $\mathrm{Pr}\left(\gamma_{i,k,l}^{\left(\eta \right)}=1|E;\theta^{\left(\eta \right)}\right)$. ${\hat{\gamma }}_{i,k,l}^{\left(\eta \right)}$ denotes the probability that observed data $e_{i,k}$ come from the *l*th Gaussian sub-model under the current model parameters, called the responsiveness of sub-model *l* to observed data $e_{i,k}$. ${\hat{\gamma }}_{i,k,l}^{\left(\eta \right)}$ can be calculated from the Bayesian rule of Equation (Equation 18).
(18)$$\begin{array}{c} {\hat{\gamma }}_{i,k,l}^{\left(\eta \right)}=\mathrm{Pr}\left(\gamma_{i,k,l}^{\left(\eta \right)}=1|E;\theta^{\left(\eta \right)}\right)=\frac{\alpha_{l}^{\left(\eta \right)}p_{e}^{\left(l\right)}\left(e_{i,k};\mu_{l}^{\left(\eta \right)},\Sigma_{l}^{\left(\eta \right)}\right)}{\sum_{l=1}^{2}\alpha_{l}^{\left(\eta \right)}p_{e}^{\left(l\right)}\left(e_{i,k};\mu_{l}^{\left(\eta \right)},\Sigma_{l}^{\left(\eta \right)}\right)} \end{array}$$

Step 2 (M-step): The maximum of function $Q\left(\theta ,\theta^{\left(\eta \right)}\right)$ is obtained from Equation (Equation 18) with θ as the vector parameter. The result of the (η+1)th iteration is
(19)$$\theta^{\left(\eta +1\right)}=\arg\max_{\theta }Q\left(\theta ,\theta^{\left(\eta \right)}\right)$$

## 4. Algorithm Implementation

The probability density function (PDF) of random variables in measurement error E is
(20)$$p_{e}\left(e\right)=\alpha_{1}N\left(e;\mu_{1},\Sigma_{1}\right)+\alpha_{2}N\left(e;\mu_{2},\Sigma_{2}\right)$$

The more appropriate initial vector parameter $\theta^{\left(0\right)}={\left[\alpha_{1}^{\left(0\right)},\alpha_{2}^{\left(0\right)},\mu_{1}^{\left(0\right)},\Sigma_{1}^{\left(0\right)},\mu_{2}^{\left(0\right)},\Sigma_{2}^{\left(0\right)}\right]}^{T}$ obtained according to the k-means++ algorithm was used for the first iteration of the EM algorithm. The cost function in (Equation 17) can be simplified as
(21)$$\Lambda^{\left(\eta \right)}\left(\theta \right)=\sum_{i=1}^{N}\sum_{k=1}^{K}\sum_{l=1}^{2}\ln\left[\alpha_{l}p_{e}^{\left(l\right)}\left(e_{i,k};\mu_{l},\Sigma_{l}\right)\right]{\hat{\gamma }}_{i,k,l}^{\left(\eta \right)}$$

In order to maximize the GMM with parameter $\Lambda^{\left(\eta \right)}\left(\theta \right)$, we can solve
(22)$$\frac{\partial }{\partial \alpha_{l}}\left[\Lambda^{\left(\eta \right)}\left(\theta \right)+\lambda \left(\sum_{l}^{2}\alpha_{l}-1\right)\right]=0$$
(23)$$\frac{\partial }{\partial \mu_{l}}\left[\Lambda^{\left(\eta \right)}\left(\theta \right)\right]=0$$
(24)$$\frac{\partial }{\partial \Sigma_{l}}\left[\Lambda^{\left(\eta \right)}\left(\theta \right)\right]=0$$
where λ in (Equation 22) is a Lagrange multiplier. In (Equation 24), $\theta ={\left[\alpha_{1}^{\left(\eta \right)},\alpha_{2}^{\left(\eta \right)},\mu_{1}^{\left(\eta +1\right)},\Sigma_{1}^{\left(\eta \right)},\mu_{2}^{\left(\eta +1\right)},\Sigma_{2}^{\left(\eta \right)}\right]}^{T}$. Meanwhile, the solutions of the equations are all in closed form, and the result is
(25)$$\alpha_{l}^{\left(\eta +1\right)}=\frac{\sum_{i=1}^{N}\sum_{k=1}^{K}{\hat{\gamma }}_{i,k,l}^{\left(\eta \right)}}{NK}$$
(26)$$\mu_{l}^{\left(\eta +1\right)}=\frac{\sum_{i=1}^{N}\sum_{k=1}^{K}e_{i,k}{\hat{\gamma }}_{i,k,l}^{\left(\eta \right)}}{\sum_{i=1}^{N}\sum_{k=1}^{K}{\hat{\gamma }}_{i,k,l}^{\left(\eta \right)}}$$
(27)$$\Sigma_{l}^{\left(\eta +1\right)}=\frac{\sum_{i=1}^{N}\sum_{k=1}^{K}{\left(e_{i,k}-\mu_{l}^{\left(\eta +1\right)}\right)}^{T}\left(e_{i,k}-\mu_{l}^{\left(\eta +1\right)}\right){\hat{\gamma }}_{i,k,l}^{\left(\eta \right)}}{\sum_{i=1}^{N}\sum_{k=1}^{K}{\hat{\gamma }}_{i,k,l}^{\left(\eta \right)}}$$

The above calculations are repeated until the log-likelihood function value no longer changes significantly. By rounding the final data ${\hat{\gamma }}_{i,k,l}^{\left(\eta +1\right)}$ of the hidden variable, we obtain the complete data set $\left\{E,\gamma \right\}$ and the vector parameter θ of the GMM.

Thus, the pseudo-algorithm of the joint use of k-means++ algorithm and EM algorithm for parameter estimation of GMM is shown in Algorithm 1.
**Algorithm 1** Joint k-means++ and EM algorithms for estimating parameters of GMM.**Input:**Y and Z. For each dataset with i=1,2,…,N, k=1,2,…,K. **Initialize:** Iteration index *n* = 0 for k-means++ algorithm; the EM algorithm’s iteration index η = 0; convergence tolerance is Δ; and maximum iteration number is Nitrmax. **K-means++ algorithm loop:** (1) A sample point is randomly selected as the initial cluster center c1(0), and then the second cluster center c2(0) is selected according to the roulette wheel selection. (2) Update γ(n) according to Equation (Equation 15), and then reclassify the sample points. (3) Update Cluster Center cln+1 according to Equation (Equation 16). (4) If the convergence condition cln+1=cln is satisfied, the k-means++ algorithm is terminated. Otherwise, set n←n+1 and return to (2). **Get the initial estimation parameters:** (1) αl0=∑γi,k,ln+1. (2) μl0=cln+1. (3) Σl0=varE;γi,k,ln=1. **EM algorithm loop:** (1) Update γ^η according to Equation (Equation 18). (2) Parameters αlη+1, μlη+1, Σlη+1 are updated according to Equations (Equation 25)–(Equation 27). (3) If the convergence condition LIθη+1;E−LIθη;E≤Δ or η+1=Nitrmax is satisfied, the EM algorithm is terminated. Otherwise, set η←η+1 and return to (1).**Output:**
E,γη+1 and θη+1.


## 5. Algorithm Analysis

### 5.1. Convergence Analysis

The essence of using k-means++ algorithm to calculate new clustering centers is to minimize the sum of squared error (SSE) function:(28)$$J\left(c_{l}^{\left(n+1\right)}\right)=\sum_{\gamma_{i,k,l}^{\left(n+1\right)}=1}{∥e_{i,k}-c_{l}^{\left(n+1\right)}∥}^{2}$$

As can be found from the algorithm, SSE is a rigorous coordinate descent procedure. Selecting the mean of the current clustering as the new clustering center ensures that SSE will be decreased at each iteration.
(29)$$J\left(c_{l}^{\left(n+1\right)}\right)\leq J\left(c_{l}^{\left(n\right)}\right)$$

Since SSE is monotonically decreasing and has a lower bound, the optimal solution $c_{l}$ that converges SSE to the minimum can finally be obtained.

For any Gaussian distribution parameter vector $\theta^{\left(\eta \right)}$ in the EM algorithm’s parameter space, updating $\alpha_{1}^{\left(\eta +1\right)}$, $\alpha_{2}^{\left(\eta +1\right)}$, $\mu_{1}^{\left(\eta +1\right)}$, $\Sigma_{1}^{\left(\eta +1\right)}$, $\mu_{2}^{\left(\eta +1\right)}$, $\Sigma_{2}^{\left(\eta +1\right)}$ is easily verified via the following relationship [37,38]:(30)$$Q\left(\theta^{\left(\eta +1\right)},\theta^{\left(\eta \right)}\right)\geq Q\left(\theta^{\left(\eta \right)},\theta^{\left(\eta \right)}\right)$$

Based on the monotonicity of the log-likelihood function $Q\left(\theta ,\theta^{\left(\eta \right)}\right)$ for complete data and the boundedness of $p\left(E;\theta \right)$ in the EM algorithm, it can be proved that the proposed EM algorithm converges to a stationary point $L_{I}^{*}$ of the log-likelihood function $L_{I}\left(\theta ;E\right)$ for incomplete data.

### 5.2. Complexity Analysis

In the complexity analysis, we focused on the iterative process between the k-means++ algorithm and the EM algorithm in the estimation of parameters. Since they consume more computationally, complexity was evaluated with floating point operations (FLOPs).

We define FLOPs in relation to some basic operations as follows:(1)$\varepsilon_{add}$: FLOPs required for addition.(2)$\varepsilon_{sub}$: FLOPs required for subtraction.(3)$\varepsilon_{mul}$: FLOPs required for multiplication.(4)$\varepsilon_{div}$: FLOPs required for division.(5)$\varepsilon_{exp}$: FLOPs required for exponential.(6)$\varepsilon_{pow}$: FLOPs required for square.(7)$\varepsilon_{sqrt}$: FLOPs required for square root.(8)$\varepsilon_{com}$: FLOPs required for comparation.(9)$\varepsilon_{ass}$: FLOPs required for assignment.

Note that the FLOPs used in actual practice may differ depending on the processor.

Since both k-means++ and EM algorithms are iterative, we focused our analysis in a single iterative process. The (n+1)th iteration of the k-means++ algorithm to reclassify the dataset according to (Equation 15) requires $NK(4\varepsilon_{sub}+4\varepsilon_{pow}+2\varepsilon_{add}+1\varepsilon_{com}+2\varepsilon_{ass})$ flops, and to update the clustering center according to (Equation 16) requires $\left(3NK-5\right)\varepsilon_{add}+4\varepsilon_{div}+1\varepsilon_{sub}$. We define FL(c) as the FLOPs required to estimate cluster center c in one iteration of the k-means++ algorithm.
(31)$$\begin{array}{cc} FL\left(c\right) & =\left(5NK-5\right)\varepsilon_{add}+\left(4NK+1\right)\varepsilon_{sub}\\ \; & +4\varepsilon_{div}+4NK\varepsilon_{pow}+NK\varepsilon_{com}+2NK\varepsilon_{ass} \end{array}$$

The update of ${\hat{\gamma }}_{i,k,l}^{\left(\eta \right)}$ needs to be evaluated during the $\left(\eta +1\right)$th iteration of the EM algorithm, where
(32)$$\begin{array}{cc} \alpha_{l}p_{e}^{\left(l\right)}\left(e_{i,k};\mu_{l},\Sigma_{l}\right) & =\frac{\alpha_{l}}{2\pi {\left|\Sigma_{l}\right|}^{1/2\;}}\cdot \exp\left[-\frac{\left(e_{i,k}-\mu_{l}\right)\Sigma_{l}^{-1}{\left(e_{i,k}-\mu_{l}\right)}^{T}}{2}\right] \end{array}$$
requires $2(\left(NK+4\right)\varepsilon_{mul}+\left(2NK+1\right)\varepsilon_{sub}+\left(NK+1\right)\varepsilon_{div}+\left(NK+1\right)\varepsilon_{add}+2NK\varepsilon_{pow}+NK\varepsilon_{exp}+1\varepsilon_{sqrt})$ FLOPs. Equation (Equation 18) requires $NK(1\varepsilon_{add}+1\varepsilon_{div}+1\varepsilon_{sub})$ FLOPs. With ${\hat{\gamma }}_{i,k,l}^{\left(\eta \right)}$, we can calculate the Equations (Equation 25)–(Equation 27), which require $\left(NK-1\right)\varepsilon_{add}+1\varepsilon_{div}+1\varepsilon_{sub}$ FLOPs, $2(2\left(NK-1\right)\varepsilon_{add}+2NK\varepsilon_{mul}+2\varepsilon_{div})$ FLOPs and $2(2\left(NK-1\right)\varepsilon_{add}+2NK\varepsilon_{sub}+2NK\varepsilon_{pow}+2NK\varepsilon_{mul}+2\varepsilon_{div})$ FLOPs, respectively. We define $FL\left(\theta \right)$ as the FLOPs required to estimate θ during each EM algorithm iteration.
(33)$$\begin{array}{cc} FL\left(\theta \right) & =\left(12NK+7\right)\varepsilon_{add}+\left(9NK+3\right)\varepsilon_{sub}\\ \; & +\left(10NK+8\right)\varepsilon_{mul}+\left(3NK+11\right)\varepsilon_{div}\\ \; & +2NK\varepsilon_{exp}+4NK\varepsilon_{pow}+2\varepsilon_{sqrt} \end{array}$$

Finally, the number of iterations required to achieve convergence is assumed to be $N_{itr}^{k}$ or $N_{itr}^{EM}$ for the k-means++ and EM algorithms, respectively. Then, the FLOPs needed to ultimately estimate the vector parameter θ are approximately
(34)$$FL\approx N_{itr}^{k}\left[FL\left(c\right)\right]+N_{itr}^{EM}\left[FL\left(\theta \right)\right]$$

## 6. Simulation Analysis

To verify the feasibility of the proposed algorithm, the simulation in this paper was performed with IEEE 5-bus standard test system and IEEE 14-bus standard test system. The MATLAB R2018b software was used for simulation, and the related data in the MATPOWER 7.1 power simulation package were used for routine power flow calculation. The final operating data were used as the measurement data for the power system. The attack vector was injected into the system first, and then the k-means++ algorithm and EM algorithm were jointly used to verify the feasibility of this detection method.

### 6.1. Simulation Parameters

The related data modified from the simulation of IEEE 5-bus standard test system are shown in Table 1. The other data were unchanged. We summarize the simulation parameters that were used in the simulation in Table 2, and generated simulation data based on these parameters to test the algorithm.

### 6.2. Simulation Results

For the 600 data points shown in Figure 1, the measurement errors of some phasors begin to shift when a meter measurement in the power system is tampered with. Figure 2 shows the initial data-clustering results processed by the k-means++ algorithm. The classification results of the GMM and data obtained after the subsequent EM algorithm are shown in Figure 3, and the images of their final classification results are basically consistent with those shown in Figure 1. Figure 4 visualizes the PDF image of the measurement error distribution of GMM, and the figure shows the error offset caused by the false data.

Figure 5 shows that the sum of squared errors of the model gradually flattens out as the number of iterations monotonically changes when using the k-means++ algorithm for simulation. Figure 6 shows that with the EM algorithm, the logarithmic likelihood function values of the model gradually flatten out as the number of iterations monotonically changes. The simulation results show that both algorithms can take little time to achieve convergence.

The simulation result shows in Figure 7 that the detected false data come from the branches $I_{1-2}$ between measurement buses 1 and 2. There was one misdetected measurement datum each in branch $I_{1-5}$ and branch $I_{4-5}$.

For a changing number of measurement buses injected with false data, the average error change of vector parameter $\theta ={\left[\alpha_{1},\alpha_{2},\mu_{1},\Sigma_{1},\mu_{2},\Sigma_{2}\right]}^{T}$ in GMM obtained by the detection method in this paper is shown in Figure 8, Figure 9 and Figure 10. It can be seen that as the false data increase in number, the estimation errors of parameters $\alpha_{2}$, $\mu_{2}$ and $\Sigma_{2}$ of this algorithm decrease continuously.

As the proportion of false data in the overall data increases, the probabilities of false data detection, missed detection and false detection by this algorithm change, as shown in Figure 11. It can be seen that the detection rate of the algorithm for false data is basically above 95%, and the detection probability can be further improved to above 99% as the amount of false data increases; thus, the probabilities of false detection and missed detection are normally below 1%.

In order to further verify the rapidity of the algorithm proposed in this paper for detecting false data injection attacks, we have conducted 1000 repeated experiments. The simulated time statistic histogram and normal distribution curve obtained after 1000 repetitions of simulation experiments are shown in Figure 12. From the normal distribution curve in the graph, it can be seen that the algorithm can basically detect false data in 0.011883 s.

To verify the feasibility of the proposed algorithm, it was further tested in the IEEE 14-bus standard test system. The measurement errors of active and reactive power of the bus and transmission lines and the errors after being attacked by false data injection are shown in Table 3. The validity of the method was verified by injecting false data into arbitrarily selected measurement units. One thousand sets of quantitative measurement vectors with false data were generated as experimental data according to the Monte Carlo method.

The attack vector injected in this paper against the IEEE 14-bus system was
(35)$$a={\left[\Delta P_{3},\Delta Q_{2},\Delta Q_{3},\Delta P_{1-2},\Delta P_{2-3},\Delta P_{4-2},\Delta Q_{1-2},\Delta Q_{2-3},\Delta Q_{4-2}\right]}^{T}$$

Firstly, the measurement errors were used to detect FDIAs. The measurement errors obtained by Monte Carlo method for 1000 instances of normal data were transformed into samples that conformed to the standard normal distribution model, and the measurement error data obtained are shown in Figure 13. All the data conform to the model of standard normal distribution, and the measurement errors of the sample data are not shifted.

The results of the measurement error after injecting false data are shown in Figure 14. It can be seen in the figure that the FDIAs with Equation (Equation 35) as the attack vector made the degree of offset of the measurement error more significant. The results of clustering the measurement errors after the false data injection attack by the k-means++ algorithm are shown in Figure 15.

The data preprocessed using the k-means++ algorithm were further iteratively calculated using the EM algorithm. The final PDF image of the GMM of the measurement error was obtained as shown in Figure 16. The results of classifying the sample data of 1000 measurement vectors according to the fitted GMM are shown in Figure 17. From the figure, it can be seen that there is no influence of bias in the normal measurement data, so its error distribution is basically around zero. The data with error deviations were removed and classified by classifying the sample data. It is known that the power measurement data of $P_{3}$, $Q_{2}$, $Q_{3}$, $P_{1-2}$, $P_{2-3}$, $Q_{2-3}$, $Q_{1-2}$ and $Q_{4-2}$ in the power system were tampered with by the attacker through FDIAs. The detection of false data in the measurement data using the algorithm of this paper is shown in Figure 18. A small number of data were identified as normal data because the data in measurement units $P_{3}$, $Q_{3}$, $P_{1-2}$ and $P_{2-3}$ are more similar to the normal data.

Secondly, we detected FDIAs from the perspective of the results of state estimation. When not under attack, 100 sets were randomly selected from the 1000 sets of measurement data for state estimation. The errors of their state estimation results were transformed into samples that conformed to the model of standard normal distribution, and the obtained estimation errors are shown in Figure 19. All data conform to the model with a standard normal distribution, and none of the sample data are biased by the measurement errors.

The results of its measurement error after injecting false data are shown in Figure 20. From the figure, it can be seen that the voltage amplitude and phase angle of the state estimate of some buses are significantly shifted.

The data preprocessed by the k-means++ algorithm were further iteratively calculated using the EM algorithm, and the final PDF image of the state estimation error conforming to the GMM is shown in Figure 21. The results of classifying the sample data of 100 state variables according to the fitted GMM are shown in Figure 22. From the figure, it can be seen that the data with error deviations were removed and classified by classifying the sample data. The errors of voltage magnitude and phase angle of bus 1 and buses 4–14 are around zero, and their deviations are very small, so they basically have no impact on the power system. The results of the state estimation of bus 3 are mainly the offset of voltage amplitude, which has a mild impact on the power system. The results of the state estimation of bus 2 show large shifts in voltage magnitude and phase angle, indicating that bus 2 was the main target of the FDIAs. The detection of false data in the measurement vector using the algorithm proposed in this paper is shown in Figure 23.

## 7. Conclusions

Considering that false data injection attacks can disrupt the secure operation of smart grids, we proposed a method to detect and locate false data injection attacks in power systems using statistical learning. By combining the k-means++ algorithm with the EM algorithm, it is possible to accurately model the smart grid bus measurement data within 0.011883s. At the same time, the GMM containing the characteristic parameters of data measurement errors can be obtained. Numerical examples showed that the mathematical model obtained by this joint algorithm provides a detection probability of more than 95% for false data, and can accurately locate the measured buses that are tampered with by FDIAs.

Subsequent research can provide the best choice of GMM with different models by combining the Akaike Information Criterion (AIC), Bayesian Information Criterion (BIC), Silhouette Coefficient (SC), Calinski–Harbasz (CH) score and other methods, so as to build a more perfect model to improve the algorithm in this paper.

## Figures and Tables

**Figure 1 sensors-23-01683-f001:**
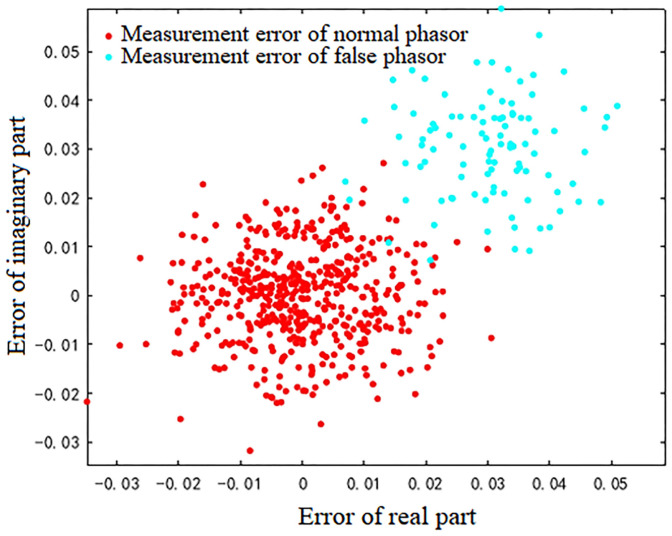
The actual distribution of phase measurement errors after injecting false data.

**Figure 2 sensors-23-01683-f002:**
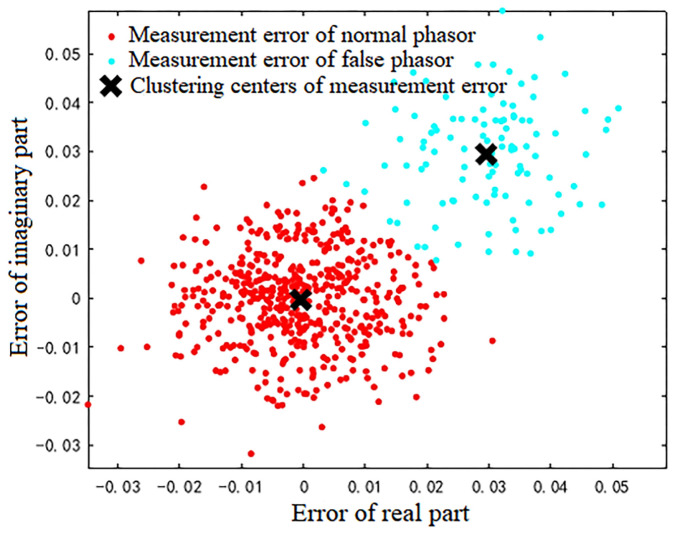
The processing results of the k-means++ algorithm.

**Figure 3 sensors-23-01683-f003:**
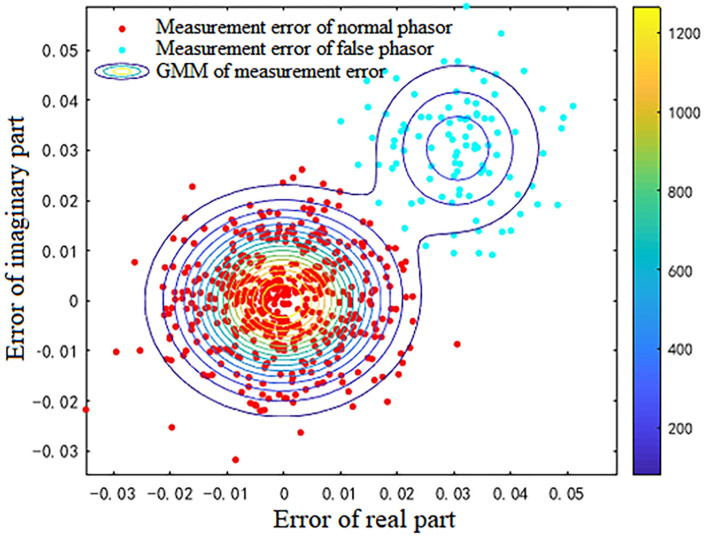
The processing results of the EM algorithm.

**Figure 4 sensors-23-01683-f004:**
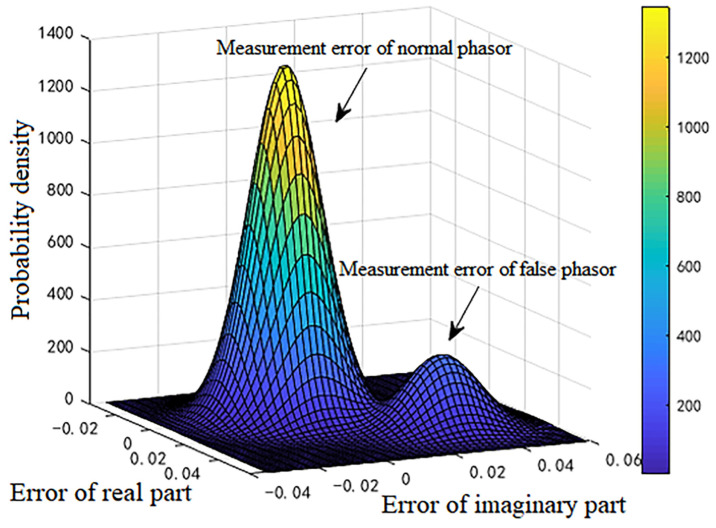
PDF of the GMM of measurement errors.

**Figure 5 sensors-23-01683-f005:**
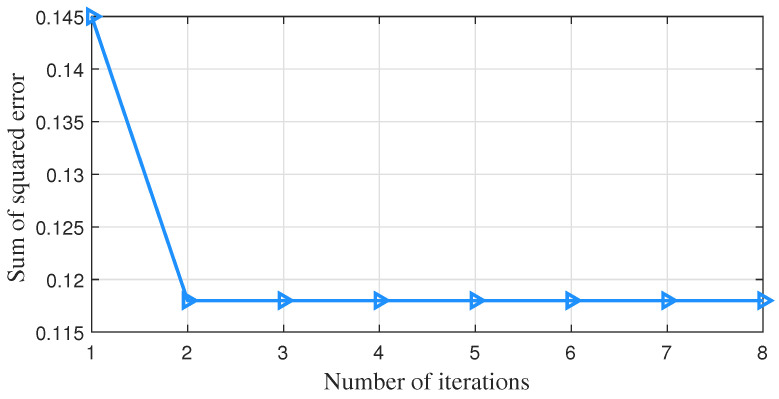
The change in the sum of the squared errors under the k-means++ algorithm.

**Figure 6 sensors-23-01683-f006:**
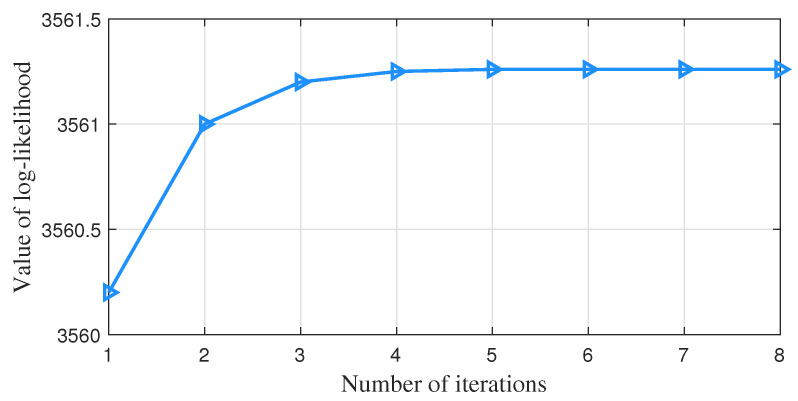
The change in the log-likelihood function value under the EM algorithm.

**Figure 7 sensors-23-01683-f007:**
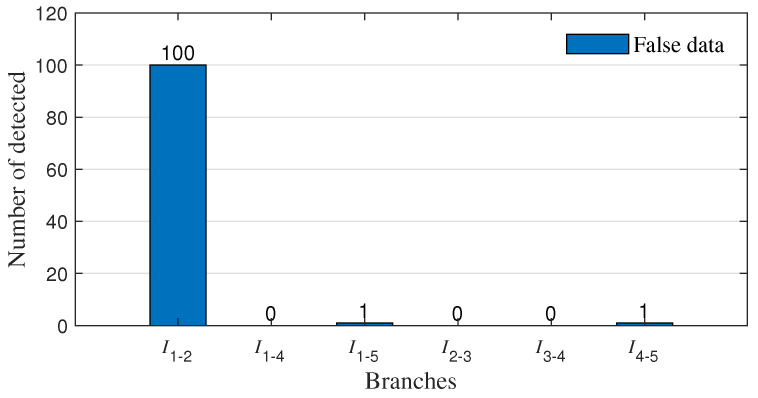
Localization of false data.

**Figure 8 sensors-23-01683-f008:**
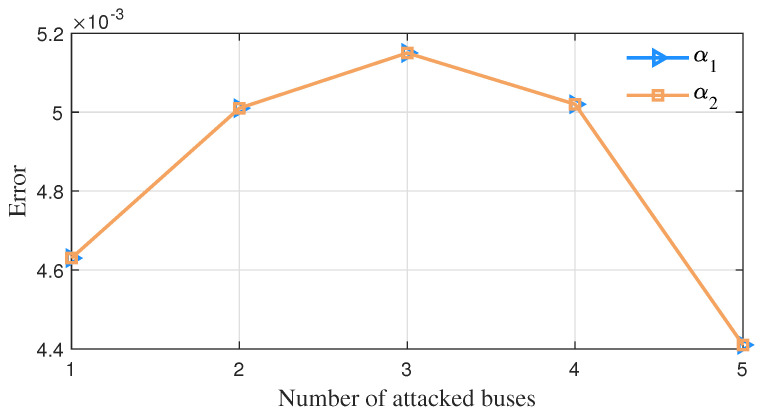
The error variation of the parameter α while the number of attacked buses varies.

**Figure 9 sensors-23-01683-f009:**
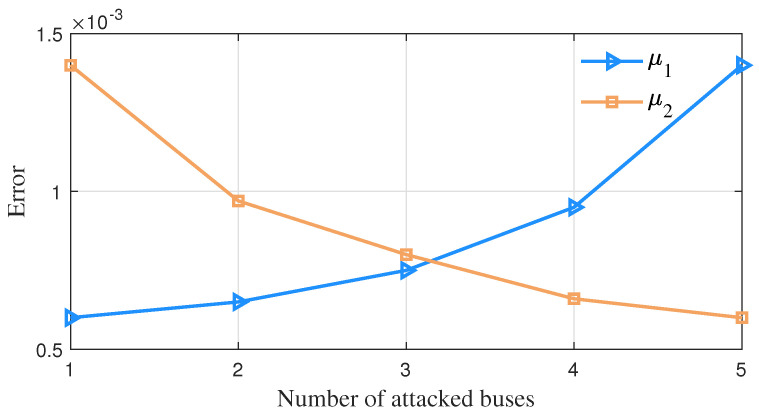
The error variation of the parameter μ while the number of attacked buses varies.

**Figure 10 sensors-23-01683-f010:**
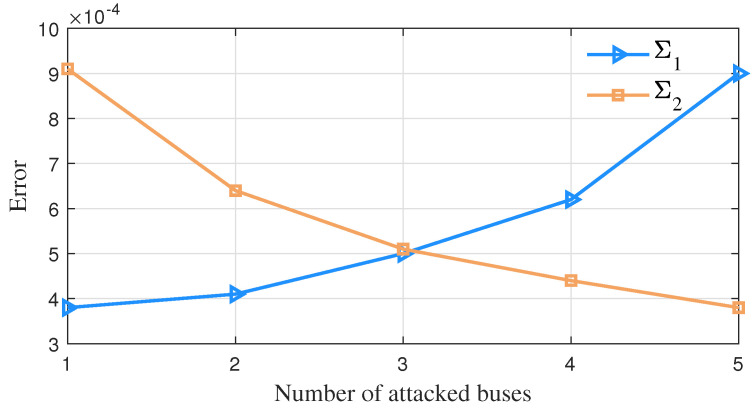
The error variation of the parameter Σ while the number of attacked buses varies.

**Figure 11 sensors-23-01683-f011:**
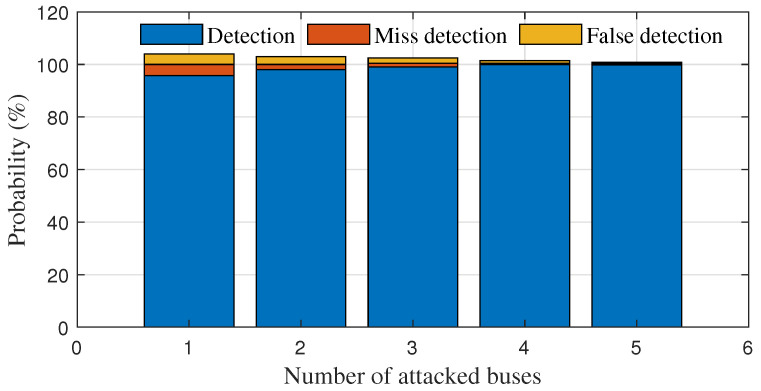
Probability of false data detection.

**Figure 12 sensors-23-01683-f012:**
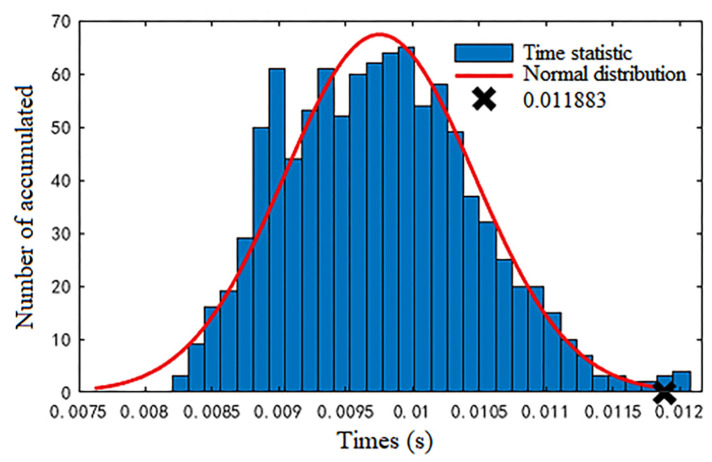
The simulation time statistics of 1000 repeated experiments and their normal distribution.

**Figure 13 sensors-23-01683-f013:**
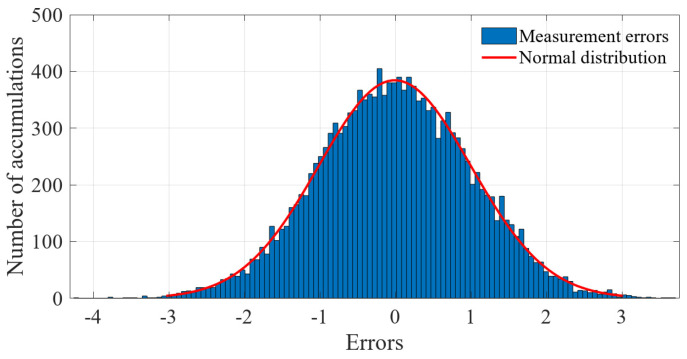
Measurement errors of normal data.

**Figure 14 sensors-23-01683-f014:**
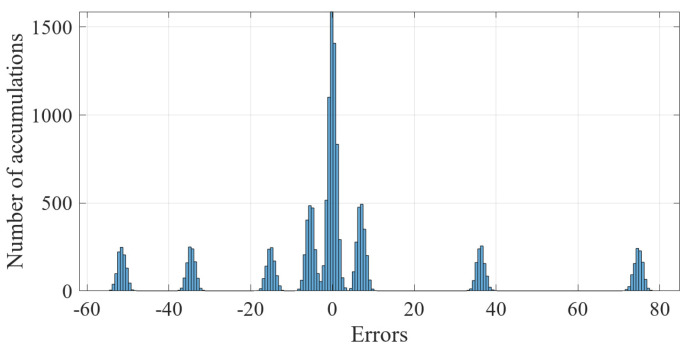
Measurement error of injecting false data.

**Figure 15 sensors-23-01683-f015:**
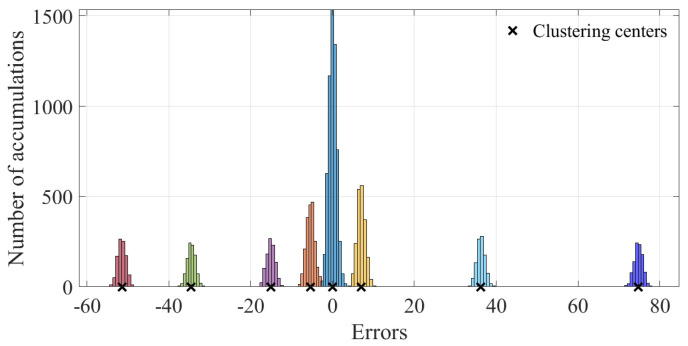
Clustering results of the k-means++ algorithm.

**Figure 16 sensors-23-01683-f016:**
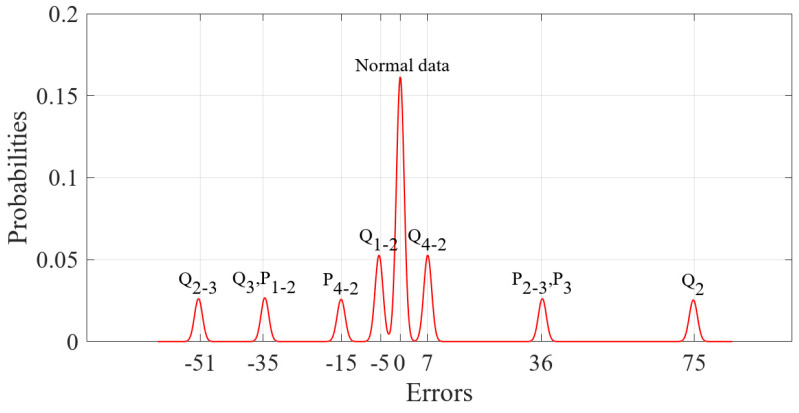
PDF of measurement errors.

**Figure 17 sensors-23-01683-f017:**
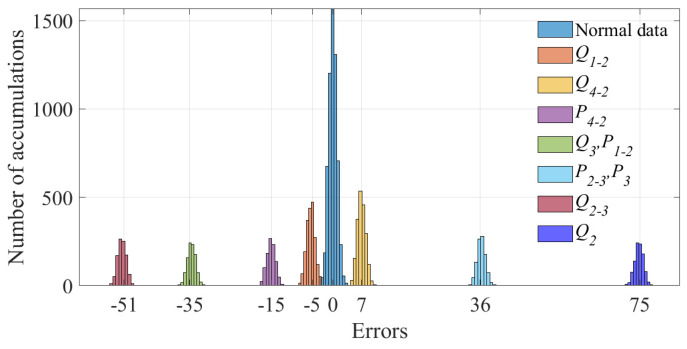
Classification results of the EM algorithm.

**Figure 18 sensors-23-01683-f018:**
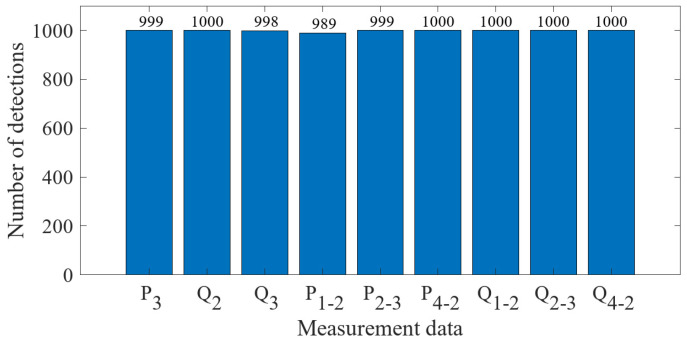
Detection results of false data.

**Figure 19 sensors-23-01683-f019:**
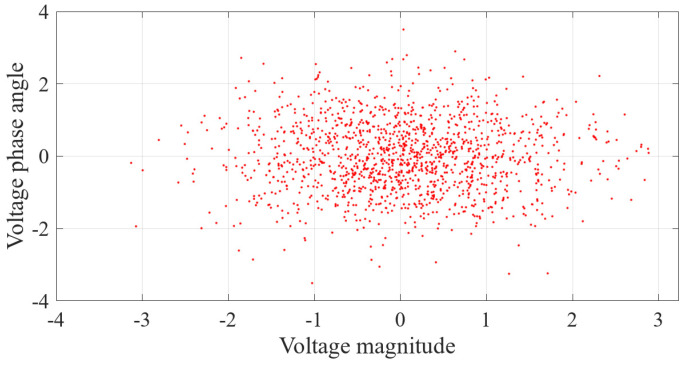
Errors of the state estimation under normal conditions.

**Figure 20 sensors-23-01683-f020:**
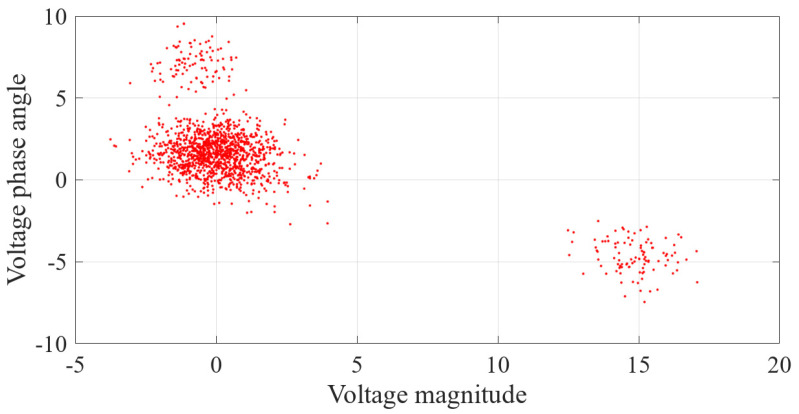
Errors of state estimation after false data injection.

**Figure 21 sensors-23-01683-f021:**
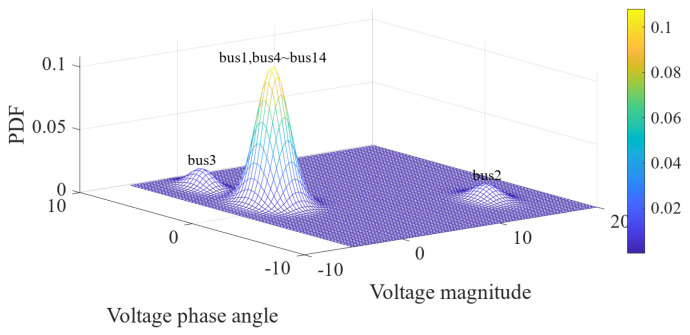
PDF of state estimation errors.

**Figure 22 sensors-23-01683-f022:**
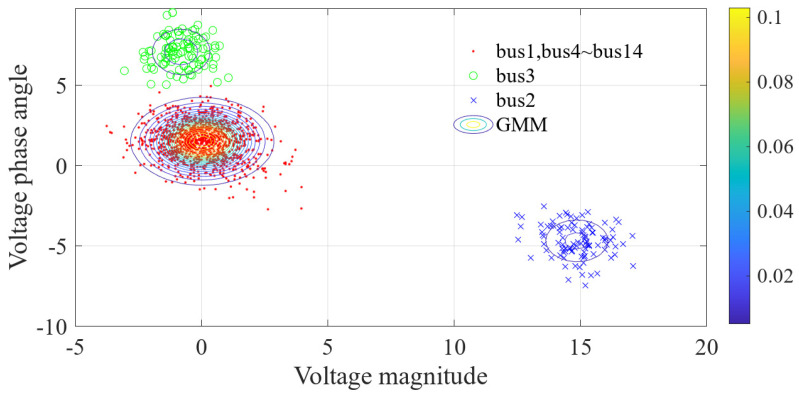
Classification results of the EM algorithm.

**Figure 23 sensors-23-01683-f023:**
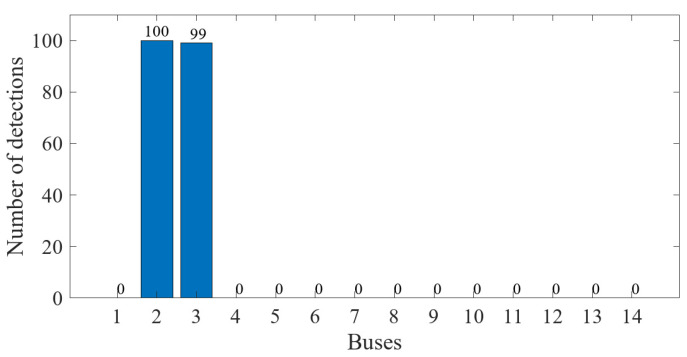
Detection results of false data.

**Table 1 sensors-23-01683-t001:** Simulation parameters.

$I_{1-2}$	Raw Data	Simulation Parameters
Amplitude/p.u.	2.5078	2.5369
Phase angle/${}^{\circ }$	−1.8803	−1.1809

**Table 2 sensors-23-01683-t002:** Simulation parameters.

Parameter	Value
*N*	6
*K*	100
$\mu_{1}$	[0 0]
$\mu_{2}$	[0.03 0.03]
σ	0.01
Δ	$10^{-6}$
$N_{itr}^{\max}$	100

**Table 3 sensors-23-01683-t003:** The measurement error before and after the power system was attacked.

Types of Measurements	Measurement Error σ before the Attack	Measurement Error σ after the Attack
$P_{i}$	0.01	0.015
$Q_{i}$	0.01	0.015
$P_{ij}$	0.008	0.012
$Q_{ij}$	0.008	0.012

## Data Availability

Not applicable.

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
