# Peer review of "Detection of False Data Injection Attacks in Smart Grids Based on Expectation Maximization"

_sensors, 2023, doi:10.3390/s23031683_

Round 1
Reviewer 1 Report
The authors present a methodology based on GMM and k-means to detect False Data Injection Attacks in Smart Grids. However, they do not highlight the novelty of the methodology.
It is necessary to rethink the inputs and strictly validate the methodology. It is not only to apply it to a five-node test system, it is necessary to analyze what happens when the number of nodes and variables increases.
The state of the art needs to be discussed in detail. No discussion of why it is necessary to use the proposed methodology compared to HMM or deep learning, for example. The authors should analyze:
"HMM-based fast detection of false data
injections in advanced metering infrastructure"
"Fault diagnosis for smart grids in pragmatic con-
ditions"
Reviewer 2 Report
This paper proposes an attack detection algorithm based on 6 statistical learning according to the different characteristic parameters of measurement error before and after tampering. The article possesses practicality and provides a technical reference for practitioners, however, I have some doubts.
How well does the EM algorithm compare to the neural network model?
The article uses a lot of equations to describe, so what form is scalar and what form is matrix? vector?
The innovation points of the article should be clearly listed item by item.
In DAGMM is also EM combined with GMM, is this article similar to this article's scheme? It is recommended to compare.
In Figure 5 and 6, we can see that the scheme of this paper only needs a few steps to get convergence, why is it so fast?
Some related papers are missing by the authors, An, P., Wang, Z., & Zhang, C. (2022). Ensemble unsupervised autoencoders and Gaussian mixture model for cyberattack detection. Information Processing & Management, 59(2), 102844.
Reviewer 3 Report
The paper “Detection of False Data Injection Attacks in Smart Grid Based on Expectation Maximization” focuses on an interesting topic that is a hot area of research these days. Protecting smart grids from false data attacks is the demand of the day. The paper has been well written. I have some questions which need to be addressed before further consideration on the final decision.
• The authors have shown the k-mean ++ algorithm for detecting the location of the attacked bus, but they did not mention the algorithm’s capability to detect the time of the attack. The authors need to explain this.
• IEEE 5-bus system has been used in this study. What is the reason for selecting the 5-bus system? Can the algorithm work well on a 14-bus system and 30 bus system? Please explain.
• The authors have shown the FDIA detection probability to be 95 %. But the recent literature shows the capability of the algorithms to achieve 100 % accuracy. For example, in https://www.mdpi.com/1424-8220/22/9/3146
• The authors need to compare the performance with recently published literature.
Round 2
Reviewer 3 Report
Paper can be accepted in its present form.